# Learning to Group: A Bottom-Up Framework for 3D Part Discovery in Unseen Categories

**Tiange Luo**
Peking University, Zhejiang Lab

**Kaichun Mo**
Stanford University

**Zhiao Huang**
UC San Diego

**Jiarui Xu**
HKUST

**Siyu Hu**
USTC

**Liwei Wang**
Peking University, BIBDR

**Hao Su**
UC San Diego

## Abstract

We address the problem of discovering 3D parts for objects in unseen categories. Being able to learn the geometry prior of parts and transfer this prior to unseen categories pose fundamental challenges on data-driven shape segmentation approaches. Formulated as a contextual bandit problem, we propose a learning-based agglomerative clustering framework which learns a grouping policy to progressively group small part proposals into bigger ones in a bottom-up fashion. At the core of our approach is to restrict the local context for extracting part-level features, which encourages the generalizability to unseen categories. On the large-scale fine-grained 3D part dataset, PartNet, we demonstrate that our method can transfer knowledge of parts learned from 3 training categories to 21 unseen testing categories without seeing any annotated samples. Quantitative comparisons against four shape segmentation baselines shows that our approach achieve the state-of-the-art performance. [Project Page]

## 1 Introduction

Perceptual grouping has been a long-standing problem in the study of vision systems (Hoffman & Richards, 1984). The process of perceptual grouping determines which regions of the visual input belong together as parts of higher-order perceptual units. Back to the 1930s, Wertheimer (1938) listed several vital factors, such as similarity, proximity, and good continuation, which lead to visual grouping. To this era of deep learning, grouping cues can be learned from massive annotated datasets. However, compared with human visual system, these learning-based segmentation algorithms are far inferior for objects from unknown categories.

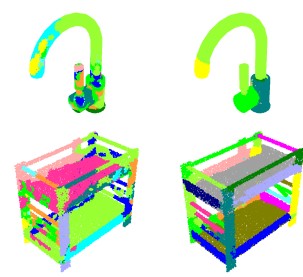

Figure 1: Shape Segmentation Results on unseen categories. Left column shows the results of SOTA deep-learning method with using global contextual information and right is ours.

We are interested in attacking a specific problem of this kind — zero-shot part discovery for 3D shapes. We choose to study the zero-shot learning problem on 3D shape data instead of 2D image data, because part-level similarity across object categories in 3D is more salient and less affected by various distortions introduced in the imaging process.

To motive our approach, we first review the key idea and limitation of existing 3D part segmentation methods. With the power of big data, deep neural networks that learn data-driven features to segment shape parts, such as (Kalogerakis et al., 2010; Graham et al., 2018; Mo et al., 2019b), have demonstrated the state-of-the-art performance on many shape segmentation benchmarks (Yi et al., 2016; Mo et al., 2019b). These networks usually have large receptive fields that cover the whole input shape, so that global context can be leveraged to improve the recognition of part semantics and shape structures. While learning such features leads to superior performance on the training

---

To whom correspondence may be addressed. Email: tiange.cs@gmail.com

categories, they often fail miserably on unseen categories (Figure 1) due to the difference of global shapes.

On the contrary, classical shape segmentation methods, such as (Kaick et al., 2014) that use manually designed features with relatively local context, can often perform much better on unseen object categories, although they tend to give inferior segmentation results on training categories (Table 1). In fact, many globally different shapes share similar part-level structures. For example, airplanes, cars, and swivel chairs all have wheels, even though their global geometries are totally different. Having learned the geometry of wheels from airplanes should help recognize wheels for cars and swivel chairs.

In this paper, we aim to invent a learning-based framework that will by design avoid using excessive context information that hurts cross-category generalization. We start from proposing a pool of superpixel-like sub-parts for each shape. Then, we learn a grouping policy that seeks to progressively group sub-parts and gradually increase recognition context. What lies in the heart of our algorithm is to learn a function to assess whether two parts should be grouped. Different from prior deep segmentation work that learns point features for segmentation mask prediction, our formulation essentially learns part-level features. Borrowing ideas from Reinforcement Learning (RL), we formalize the process as a contextual bandit problem and train a local grouping policy to iteratively pick a pair of most promising sub-parts for grouping. In this way, we restrict that our features only convey information within the local context of a part. Our *learning-based agglomerative clustering* framework deviates drastically from the prevailing deep segmentation pipelines and makes one step towards generalizable part discovery in unseen object categories.

To summarize, we make the following contributions:

- We formulate the task of zero-shot part discovery on the large-scale fine-grained 3D part dataset PartNet (Mo et al., 2019b);
- We propose a learning-based agglomerative clustering framework that learns to group for proposing parts from training categories and generalizes to unseen categories;
- We quantitatively compare our approach to four baseline methods and demonstrate the state-of-the-art results for part discovery in unseen categories.

## 2 RELATED WORK

Shape segmentation has been a classic and fundamental problem in computer vision and graphics. Dated back to 1990s, researchers have started to design heuristic geometric criterion for segmenting 3D meshes, including methods based on morphological watersheds (Mangan & Whitaker, 1999), K-means (Shlafman et al., 2002), core extraction (Katz et al., 2005), graph cuts (Golovinskiy & Funkhouser, 2008), random walks (Lai et al., 2008), spectral clustering (Liu & Zhang, 2004) and primitive fitting (Attene et al., 2006a), to name a few. See Attene et al. (2006b); Shamir (2008); Chen et al. (2009) for more comprehensive surveys on mesh segmentation. Many papers study mesh co-segmentation that discover consistent part segmentation over a collection of shapes (Golovinskiy & Funkhouser, 2009; Huang et al., 2011; Sidi et al., 2011; Hu et al., 2012; Wang et al., 2012; Van Kaick et al., 2013). Our approach takes point clouds as inputs as they are closer to the real-world scanners. Different from meshes, point cloud data lacks the local vertex normal and connectivity information. Kaick et al. (2014) segments point cloud shapes under the part convexity constraints. Our work learns shared part priors from training categories and thus can adapt to different segmentation granularity required by different end-stream tasks.

In recent years, with the increasing availability of annotated shape segmentation datasets (Chen et al., 2009; Yi et al., 2016; Mo et al., 2019b), many supervised learning approaches succeed in refreshing the state-of-the-arts. Kalogerakis et al. (2010); Guo et al. (2015); Wang et al. (2018a) learn to label mesh faces with semantic labels defined by human. See Xu et al. (2016) for a recent survey. More recent works propose novel 3D deep network architectures segmenting shapes represented as 2D images (Kalogerakis et al., 2017), 3D voxels (Maturana & Scherer, 2015), sparse volumetric representations (Klokov & Lempitsky, 2017; Riegler et al., 2017; Wang et al., 2017; Graham et al., 2018), point clouds (Qi et al., 2017a;b; Wang et al., 2018b; Yi et al., 2019b) and graph-based representations (Yi et al., 2017). These methods take advantage of sufficient training samples of seen categories and demonstrate appealing performance for shape segmentation. However, they often perform much worse when testing on unseen categories, as the networks overfit their weights to the

global shape context in training categories. Our work focus on learning context-free part knowledges and perform part discovery in a zero-shot setting on unseen object classes.

There are also a few relevant works trying to reduce supervisions for shape part segmentation. Makadia & Yumer (2014) learns from sparsely labeled data that only one vertex per part is given the ground-truth. Yi et al. (2016) proposes an active learning framework to propogate part labels from a selected sets of shapes with human labeling. Lv et al. (2012) proposes a semi-supervised Conditional Random Field (CRF) optimization model for mesh segmentation. Shu et al. (2016) proposes an unsupervised learning method for learning features to group superpixels on meshes. Our work processes point cloud data and focus on a zero-shot setting, while part knowledge can be learned from training categories and transferred to unseen categories.

Our work is also related to many recent research studying learning based bottom-up methods for 2D instance segmentation. These methods learn an per-pixel embedding and utilize a clustering algorithm (Newell et al., 2017; Fathi et al., 2017) as post-process or integrating a recurrent mean-shift module (Kong & Fowlkes, 2018) to generate final instances. Bai & Urtasun (2017) predicts the energy of the watershed transform and Liu et al. (2017) predicts object breakpoints and use a cascade of networks to group the pixels into lines, components and objects sequentially. Our work is significant different from previous methods as our method does not rely on an fully convolutional neural network to process the whole scene. Our work can generalize better to unseen categories as our method reduces the influences of context.

Some works in the 3D domain try to use part-level information are also related to our work (Yi et al., 2019a; Achlioptas et al., 2019; Mo et al., 2019a). Achlioptas et al. (2019) shows that the shared part-based structure of objects enables zero-shot 3D recognition based on language. To reduce the overfitting of global contextual information, our approach would exploit the part prior encoded in the dataset and involve only part-level inductive biases.

## 3 PROBLEM FORMULATION

We consider the task of zero-shot shape part discovery on 3D point clouds in unseen object categories. For a 3D shape $S$ (*e.g.* a 3D chair model), we consider the point cloud $C_S = \{p_1, p_2, \cdots, p_N\}$ sampled from the surface of the 3D model. A part $P_i = \{p_{i_1}, p_{i_2}, \cdots, p_{i_t}\} \subseteq C_S$ defines a group of points that has certain interesting semantics for some specific downstream task. A set of part proposal $\mathcal{P}_S = \{P_1, P_2, \cdots, P_S\}$ comprises of several interesting part regions on $S$ that are useful for various tasks. The task of shape part discovery on point clouds is to produce $\mathcal{P}_S^{pred}$ for each input shape point cloud $C_S$. Ground-truth proposal set $\mathcal{P}_S^{gt}$ is a manually labeled set of parts that are useful for some human-defined downstream tasks. A good algorithm should predict $\mathcal{P}_S^{pred}$ such that $\mathcal{P}_S^{gt} \subseteq \mathcal{P}_S^{pred}$ within an upper-bound limit of part numbers $M$.

A category of shapes $T = \{S_1, S_2, \cdots\}$ gathers all shapes that belong to one semantic category. For example, $T_{chair}$ includes all chair 3D models in a dataset. Zero-shot shape part discovery considers two sets of object categories $\mathcal{T}_{train} = \{T_1, T_2, \cdots, T_u\}$ and $\mathcal{T}_{test} = \{T_{u+1}, T_{u+2}, \cdots, T_v\}$, where $T_i \cap T_j = \emptyset$ for any $i \neq j$. For each shape $S \in T \in \mathcal{T}_{train}$, a manually labeled part proposal subset $\mathcal{P}_S^{gt} \subseteq \mathcal{P}_S$ is given for algorithms to use. It provides algorithms an opportunity to develop the concept of parts in the training categories. No ground-truth part proposals are provided for shapes in testing categories $\mathcal{T}_{test}$. Algorithms are expected to predict $\mathcal{P}_S^{pred}$ for any shape $S \in T \in \mathcal{T}_{test}$.

## 4 METHOD

Our method starts with proposing a set of small superpixel-like (Ren & Malik, 2003) sub-parts of the given shape. We refer readers to Appendix A for more details of our sub-part proposing method. Given a set of sub-parts, our method iteratively groups together the sub-parts belonging to the same parts in ground-truth and produce larger sub-parts, until no sub-part can further group each other. The remaining sub-parts in the final stage become a pool of part proposals for the input shape.

Our perceptual grouping process is a sequential decision process. We formulate the perceptual grouping process as a contextual bandit (one-step Markov Decision Process) (Langford & Zhang, 2007). In each iteration, we use a policy network to select a pair of sub-parts and send it to the

verification network to verify whether we should group the selected pair of sub-parts. If yes, we group the selected pair of sub-parts into a larger sub-part. Otherwise, we will not consider this pair in the latter grouping process. Our policy network is composed of two sub-modules: a purity module and a rectification module. The purity module inputs unary information and measures how likely a pair of sub-parts belong to the same part in ground-truth after grouping and the rectification module inputs binary information and further decides the pair to select. We describe more technical network design choices in Section 4.1. To train the entire pipeline, we borrow the on-policy training scheme from Reinforcement Learning (RL) to train these networks, in order to match the data distribution during training and inference stages, as described in Section 4.2.

## 4.1 MODULE DESIGNS

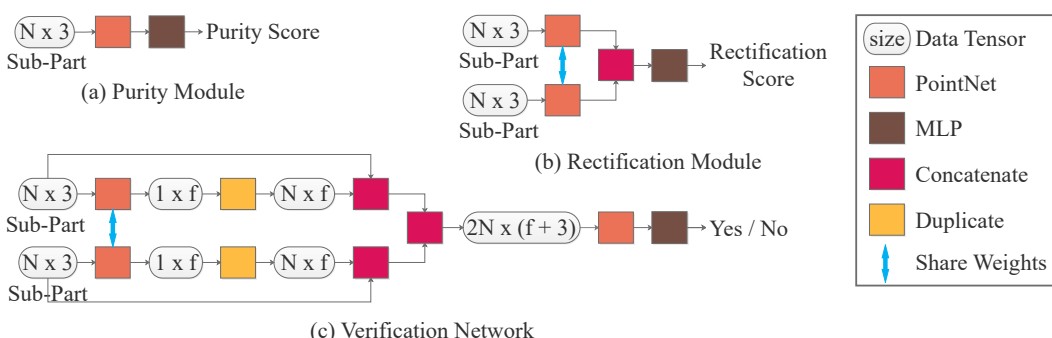

Figure 2: Network Architectures for three network modules. N is the point number of input sampled from a sub-part. f is the dimension of feature.

**Purity Module:** A pair of sub-parts $\{P_i, P_j\}$ that belong to the same ground-truth part should group together. We define *purity score* $U(P)$ for a sub-part $P$ as the maximum ratio of the intersection of $P$ with the ground-truth parts $\{P_i^{gt}\}$. More formally,

$$U(P) = \max_{P_i^{gt}} \frac{\sum_p I[p \in P]I[p \in P_i^{gt}]}{\sum_p I[p \in P]} \tag{1}$$

where $p$ enumerates all points in the shape point cloud and $I$ is the indicator function.

We train a purity module to predict the purity score. It employs a PointNet that takes as input a merged sub-part $P_{ij} = P_i \cup P_j$ and predicts the purity score. Figure 2 (a) shows the architecture.

**Rectification Module:** We observe that a purity module is not enough to fully select the best pair of sub-parts to group in practice. For example, when a large sub-part tries to group with a small one from a different ground-truth part, the part geometry of the grouping outcome is primarily dominated by the large sub-part, and thus the purity module tends to produce a high purity score, which results in selecting a pair that should not be grouped. To address this issue, we consider learning a rectification module to correct the failure case given by the purity module.

We design the rectification module as in Figure 2 (b). The rectification module takes two sub-parts separately as inputs, extracts features using a shared PointNet, concatenates the two part features and outputs a real-valued *rectification score* $R(P)$, based purely on local information. Different from the purity module that takes the grouped subpart as input, the rectification module explicitly takes two sub-parts as inputs in order to compare the two sub-part features for decision making.

**Policy Network:** We define *policy score* by making the product of *purity score* and *rectification score*. We define the policy $\pi(P_i, P_j | \mathcal{P})$ as a distribution over all possible pairs characterized by a softmax layer as shown in line 6 of Algorithm 1. The goal of the policy is to maximize the objective

$$\underset{\pi}{\text{maximize}}\ \mathbb{E}_{a \sim \pi(P_i, P_j | \mathcal{P})}\left[M(a)\right].$$

The reward, or the merge-ability score $M(P_i, P_j)$ defines whether we could group two sub-parts $P_i$ and $P_j$. To compute the reward $M(P_i, P_j)$: we first calculate the instance label of the corresponding

---

**Algorithm 1** Sub-Part Pair Selection and Grouping.

---

**Input:** A sub-parts pool $\mathcal{P} = \{P_i\}_{i \leq n}$
**Input:** Purity module $U$; Rectification module $R$; Verification network $V$
  1: **for** $i, j \leq n$ **do**
  2:       Group two shapes: $P'_{ij} \leftarrow \{P_i \cup P_j\}$
  3:       Calculate the purity score $u_{i,j} \leftarrow U(P'_{ij})$
  4:       Calculate the rectification score $r_{ij} \leftarrow R(P_i, P_j)$
  5: **end for**
  6: Calculate policy $\pi(P_i, P_j) \leftarrow \frac{e^{r_{ij} u_{ij}}}{\sum_{i,j} e^{r_{ij} u_{ij}}}$
  7: **if** isTraining **then**
  8:       Sample pair $P_i, P_j \sim \pi(P_i, P_j)$
  9: **else**
 10:       Select the $P_i, P_j = \arg \max \pi(P_i, P_j)$
 11: **end if**
 12: **if** $V(P_i, P_j)$ is True **then**
 13:       Delete $P_i, P_j$ from the pool
 14:       Add $P'_{ij}$ into the pool
 15: **end if**

---

ground-truth part for sub-parts $P_i, P_j$ as $l_i$ and $l_j$. We set $M(P_i, P_j)$ to be one if the two sub-parts have the same instance label and the purity scores of two sub-parts are greater than $0.8$.

**Verification Network:** Since the policy scores sum to one overall pairs of sub-parts, there is no explicit signal from the policy network on whether the pair should be grouped. We train a separate verification network that is specialized to verify whether we should group the selected pair. Here also exists a cascaded structure where the verification network will focus on the pairs selected by the policy network and make a double verification.

The verification network takes a pair of shape as input and outputs values from zero to one after a Sigmoid layer. Figure 2 (c) illustrates the network architecture: a PointNet first extracts the part feature for each sub-part, then two sub-part point clouds are augmented with the extracted part features and concatenated together to pass through another PointNet to obtain the final score. Notice that our design of the verification network is a combination of the purity module and rectification module. We want to extract both the input sub-part features and the part feature after grouping.

## 4.2 NETWORK TRAINING

In this section, we illustrate how to train the two networks jointly as an entire pipeline. We use Reinforcement Learning (RL) on-policy training and borrow the standard RL training techniques, such as epsilon-greedy exploration and replay buffer sampling. We also discuss the detailed loss designs for training the policy network and the verification network.

**RL On-policy Training**   Borrowing ideas from the field of Reinforcement Learning (RL), we train the policy network and the verification network in an on-policy fashion. On-policy training alternates between the data sampling step and the network training step. The data sampling step fixes the network parameters and then runs the inference-time pipeline to collect the grouping trajectories, including all pairs of sub-parts seen during the process and all the grouping operations taken by the pipeline. The network training step uses the trajectory data collected from the data sampling step to compute losses for different network modules and performs steps of gradient descents to update the network parameters. We fully describe the on-policy training algorithm in Algorithm 2.

We adapt epsilon-greedy strategy (Mnih et al., 2013) into the training stage. We start from involving 80% random sampling samples during inference as selected pairs and decay the ratio with 10% step size in each epoch. We find that random actions not only improve the exploration in the action space and but also serve as the data-augmentation role. The random actions collect more samples to train the networks, which improves the transfer performance in unseen categories.

Also, purely on-policy training would drop all experience but only use the data sampled by current policy. This is not data efficient, so we borrow the idea from DQN (Mnih et al., 2013) and use the replay buffer to store and utilize the experience. The replay buffer stores all the states and actions

---

**Algorithm 2** RL On-policy Training Algorithm.

---

**Input:** Purity module $U_\theta$ parameterized by $\theta$; Rectification module $\pi_\phi$ parameterized by $\phi$; Verification network $V$

1: Initialize buffer $B$ and the networks
2: **while** True **do**
3:      Sample shape $S$ and its ground truth-label $gt$.
4:      Preprocess $S$ to get a sub-parts pool $\mathcal{P} = \{P_i\}_{i \leq n}$
5:      **while** $\exists$ Groupable sub-parts **do**
6:          Select and group two sub-parts $P_i, P_j$ with Algorithm 1
7:          Store $(P_i, P_j, \mathcal{P})$ in $B$ and update sub-part pool $\mathcal{P}$
8:          Sample batch of data $(P_i^k, P_j^k, \mathcal{P}^k)_{k \leq N}$ from the buffer
9:          Set purity score $U_{gt}^k = U(P_i^k \cup P_j^k)$
10:         Set reward $M_{gt}^k = M(P_i^k, P_j^k)$
11:         Update rectification module with policy gradient:

$$\nabla_\phi \approx \frac{1}{N} \sum_{k \leq N} \nabla \log \pi_\phi(P_i^k, P_j^k | \mathcal{P}^k) M_{gt}^k$$

12:         Update purity module by minimizing the $l_2$ loss with purity score $U_{gt}^k$:

$$\mathcal{L}_{\text{purity}} = \frac{1}{N} \sum_{k \leq N} \|U_\theta(P_{ij}^k) - U_{gt}^k\|_2^2$$

13:         Update verification network by minimizing the cross entropy loss :

$$\mathcal{L}_{\text{verification}} = \frac{1}{N} \sum_{k \leq N} M_{gt}^k \log V(P_i^k, P_j^k) + (1 - M_{gt}^k) \log \left(1 - V(P_i^k, P_j^k)\right)$$

14:     **end while**
15: **end while**

---

during the inference stage. When updating the policy networks, we sample a batch of transitions, *i.e.* , the grouped sub-parts, and the sub-part pools when the algorithm groups the sub-parts from the replay buffer. The batch data is used to compute losses and gradients to update the two networks.

**Training Losses**   As shown in Algorithm 2, to train the networks, we sample a batch of data $(P_i^k, P_j^k, \mathcal{P}^k)_{k \leq N}$ from the replay buffer, where $P_i^k, P_j^k$ is the grouped pair and $\mathcal{P}^k$ is the corresponding sub-parts pool. We first calculate the reward $M_{gt}^k$ and ground-truth purity score $U_{gt}^k$ for each data in the batch. For updating the rectification module, we fix the purity module and calculate the policy gradient (Sutton et al., 2000) of the policy network with the reward $M_{gt}^k$ shown in line 11. As the rectification module is a part of the policy network, the gradient will update the rectification module by backpropagation. We then use the $l_2$ loss in line 12 to train the purity module and use the cross entropy loss in line 13 to train the verification network.

## 5 EXPERIMENTS AND ANALYSIS

In this section, we conduct quantitative evaluations of our proposed framework and present extensive comparisons to four previous state-of-the-art shape segmentation methods using PartNet dataset (Mo et al., 2019b) in zero-shot part discovery setting. We also show a diagnostic analysis of how the discovered part knowledge transfers across different object categories and how involving more context will affect cross-category generalization. Finally, we perform ablation studies to validate the design of the policy network.

### 5.1 DATASET AND EVALUATION

We use the recently proposed PartNet dataset (Mo et al., 2019b) as the main testbed. PartNet provides fine-grained, hierarchical and instance-level part annotations for 26,671 3D models from 24 object categories. PartNet defines up to three levels of non-overlapping part segmentation for each

object category, from coarse-grained parts (*e.g.* chair back, chair base) to fine-grained ones (*e.g.* chair back vertical bar, swivel chair wheel). Unless otherwise noticed, we use 3 categories (*i.e.* Chair, Lamp, and Storage Furniture)[1] for training and take the rest 21 categories as unseen categories for testing.

In zero-shot part discovery setting, we aim to propose parts that are useful under various different use cases. PartNet provides multi-level human-defined semantic parts that can serve as a sub-sampled pool of interesting parts. Thus, we adopt Mean Recall (Hosang et al., 2015; Sung et al., 2018) as the evaluation metric to measure how the predicted part pool covers the PartNet-defined parts. To elaborate on the calculation of Mean Recall, we first define $R_t$ as the fraction of ground-truth parts that have Intersection-over-Union (IoU) over $t$ with any predicted part. Mean Recall is then defined as the average values of $R_t$'s where $t$ varies from 0.5 to 0.95 with 0.05 as a step size.

## 5.2 BASELINE METHODS

We compare our approach to four previous state-of-the-art methods as follows:

- **PartNet-InsSeg**: Mo et al. (2019b) proposed a part instance segmentation network that employs a PointNet++ (Qi et al., 2017b) as the backbone that takes as input the whole shape point cloud and directly predicts multiple part instance masks. The method is a top-down label-prediction method that uses the global shape information.
- **SGPN**: Wang et al. (2018b) presented a learning-based bottom-up pipeline, which inputs the whole shape point cloud to extract per-point features and compute pairwise affinity matrix for point clustering. The global context is involved in learning the features.
- **GSPN**: Yi et al. (2019b) introduced a deep region-based method that learns generative models for part proposals. The method proposes local bounding boxes but still uses global-aware features for predicting boxes and segmenting parts inside local boxes.
- **WCSeg**: Kaick et al. (2014) is a non-learning based method based on the convexity assumption of parts. The method leverages hand-engineered heuristics relying on local statistics to segment shapes, thus is more agnostic to the object categories.

All the three deep learning-based methods take advantage of the global shape context to achieve state-of-the-art shape part segmentation results on PartNet. However, these networks are prone to over-fitting to training categories and have a hard time transferring part knowledge to unseen categories. WCSeg, as a non-learning based method, demonstrates good generalization capability to unseen categories, but is limited by the part convexity assumption.

| | Seen Category | | | | | Unseen Category | | | | | | | | |
|---|---|---|---|---|---|---|---|---|---|---|---|---|---|---|
| | 🪑 | 💡 | 🗄 | Avg | WAvg | 👜 | 🛏 | 🍾 | 🥣 | 🕐 | 📟 | 🖥 | 🚪 | 🎧 |
| PartNet | **55.3** | 50.3 | **23.4** | 43 | **47.4** | 18.2 | 9.7 | 40.7 | **73.5** | 30.3 | 29.3 | 43.6 | 32.1 | 16.5 |
| SGPN | 42.2 | 44.2 | 11.5 | 32.6 | 36 | 21.4 | 7 | 46.7 | 53.3 | 27.7 | 8.7 | 34.8 | 28.9 | 25.5 |
| GSPN | 39.7 | 43.7 | 14.4 | 32.6 | 35 | 34.4 | 8.4 | 46.9 | 72.8 | 40.6 | **40.6** | 57.8 | 36.7 | 28.4 |
| WCSeg | 33.1 | 56.8 | 3.2 | 31 | 31.4 | **41.9** | 8.6 | **56.3** | 69.3 | 34.2 | 27.6 | 59.7 | 30.2 | **37.3** |
| Our | 50.6 | **57** | 21.7 | **43.1** | 45.6 | 41.6 | **10.4** | 49.2 | 72.2 | **42.4** | 31.2 | **67** | **37.2** | 33.1 |

| | Unseen Category | | | | | | | | | | | | | |
|---|---|---|---|---|---|---|---|---|---|---|---|---|---|---|
| | 🚰 | 🎩 | ⌨ | ✏ | 💻 | 📱 | 🗑 | 🗄 | ✂ | 🪑 | 🗑 | 🏺 | Avg | WAvg |
| PartNet | 16.6 | **52.5** | 0.4 | 33.6 | 82.1 | 29.6 | 33 | 25 | 0.8 | 38.9 | 12.2 | 36.8 | 31.2 | 35.7 |
| SGPN | 20 | 37 | 0.4 | 31 | 67.3 | 7.2 | 13.3 | 5.9 | 6.4 | 34.8 | 7.8 | 27.5 | 24.4 | 30.8 |
| GSPN | 25.3 | 31.7 | 0.4 | 18.9 | 92.9 | **39.2** | 40.6 | 26.4 | 3.7 | 34.6 | 12.7 | 41.4 | 35.1 | 34.7 |
| WCSeg | **48.2** | 48.7 | 0.3 | **60.1** | 64.8 | 30.8 | 46 | 19.5 | **39** | 31.4 | 12.3 | 29 | 37.9 | 33.5 |
| Ours | 30.9 | 34.1 | 0.4 | 44.1 | **96.6** | 34.3 | **48.2** | **26.6** | 16.7 | **44.1** | **13** | **43.1** | **38.9** | **42.1** |

Table 1: Quantitative Evaluation. The number is the average among mean recall of three levels segmentation results in PartNet. Avg and WAvg are average among categories and weighted average among categories over shape numbers, respectively.

## 5.3 RESULTS AND ANALYSIS

We compare our proposed framework to the four baseline methods under the Mean Recall metric. For PartNet-InsSeg, SGPN, GSPN and our method, we train three networks corresponding to three

---

[1]We pick the three categories because they are big categories with several thousand models per category and provide a large variety of parts for learning.

levels of segmentation for training categories (*e.g.* Chair, Lamp, and Storage Furniture). We remove the part semantics prediction branch from the three baseline methods, as semantics are not transferable to novel testing categories. For WCSeg, point normals are required by the routine to check local patch continuity. PartNet experiments (Mo et al., 2019b) usually assume no such point normals as inputs. Thus, we approximately compute normals based on the input point clouds by reconstructing surface with ball pivoting (Bernardini et al., 1999). Then, to obtain three-levels of part proposals for WCSeg, we manually tune hyperparameters in the procedure at each level of part annotations on training categories to have the best performance on three seen categories. Since the segmentation levels for different categories may not share consistent part granularity (*e.g.* display level-2 parts may correspond to chair level-3 parts), we gather together the part proposals generated by methods at all three levels as a joint pool of proposals for evaluation on levels of unseen categories. For the proposed method, we involve limited context only on seen categories as presented in Appendix 5.5.

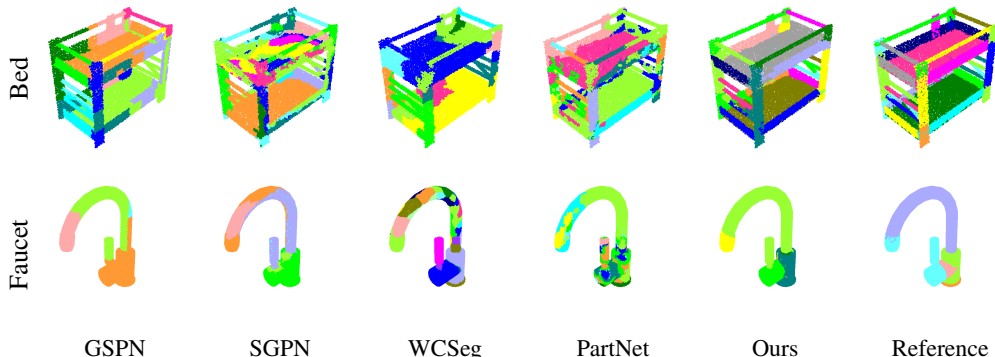

Figure 3: Qualitative results on unseen categories. More visualizations can be found in Appendix C.

We present quantitative and qualitative evaluations to baseline methods in Table 1, Figure 3 and Appendix C. For each testing category, we report the average values of Mean Recall scores at all levels. See the appendix Table 6 for detailed numbers at all levels. We observe that our approach achieves the best performance on two kinds of average among all testing novel categories.

## 5.4 PART KNOWLEDGE TRANSFER ANALYSIS

The core of our method is to learn local-context part knowledge from training categories that is able to transfer to novel unseen categories. Such learned part knowledge may also include non-transferable category-specific information, such as the part geometry and the part boundary types. Training our framework on more various object categories is beneficial to learn more generalizable knowledge that shares in common. However, due to the difficulties in acquiring human annotated fine-grained parts (*e.g.* PartNet (Mo et al., 2019b)), we can often conduct training on a few training categories. Thus, we are interested to know how to select categories to achieve the best performance in all categories.

|  | Train Category | | |
|---|---|---|---|
|  | 🪑 | 🪨 | 🛋 |
| 🪑 | 37.1 | 23.7 | 8.3 |
| 🪨 | 32.6 | 33.5 | 8.8 |
| 🛋 | 30.9 | 18.8 | 33.4 |

Table 2: Cross-validation experiments for analyzing how part knowledge transfers across category boundaries.

Different object categories have different part patterns that block part knowledge transfers across category boundaries. However, presumably, similar categories, such as tables and chairs, often share common part patterns that are easier to transfer. For example, tables and chairs are both composed of legs, surfaces, bar stretchers and wheels, which offers a good opportunity for transferring local-context part knowledge. We analyze the capability of transferring part knowledge across category boundaries under our framework. Table 2 presents experimental results of doing cross-validation using chairs, tables and lamps by training on one category and testing on another. We observe that, chairs and tables transfer part knowledge to each other as expected, while the network trained on lamps demonstrates much worse performance on generalizing to chairs and tables.

## 5.5 CONTEXT EFFECTS

In this section, we conduct experiments about the effects of involving more context for training models on seen categories and unseen categories. We add a branch to the verification network and extend it into Figure 4. This branch takes all the sub-parts where the minimum distance with the input sub-part pair $\ell_2 \leq 0.01$ as input and thus encodes more context information for decision making. Note that the involved context is still restricted in a very local region and the module can not "see" the whole shape. Now, there are two branches can be used to determine whether we should group the pair. The original binary branch is driven by purely local contextual information. The newly added one encodes more context information.

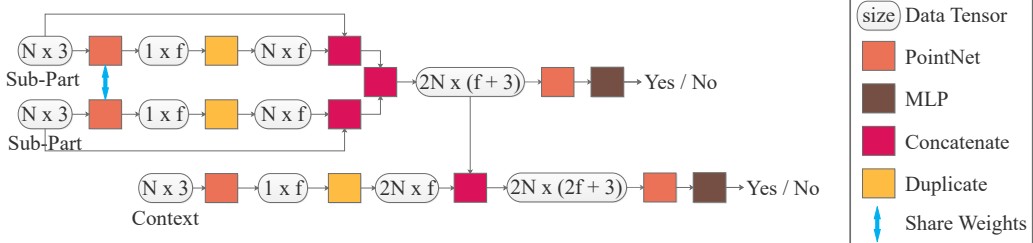

Figure 4: The extended verification network with two prediction branches. The above branch exploits pure local context. The below branch encode more context to make decisions.

To test the effectiveness of involving more context, we train the model with the extended verification network on Chair, Lamp, and Storage Furniture and test them in two ways. 1) We make decisions by only using the original binary branch. 2) We make decisions by using the original binary branch when the size of the sub-part pool is $\geq 32$ and using the newly added branch when the size of the sub-part pool is $\leq 32$. We choose 32 as an empirical threshold here based on our observation that when the size of the pool is $\leq 32$, the sub-parts in the pool will be relative large and the additional local context will help make grouping decisions in most scenarios.

From the results listed in Table 3, we can point out that the involved context helps to consistently improve the performance on seen categories, but has negative effects on most unseen categories. When the context is similar between the seen categories and unseen categories, such as patterns between Storage Furniture and Bed, the involved context can help make decisions. The phenomenon indicates a future direction that we train the model with involving more context but use them on unseen categories only when we found a similar context we have seen during training. It also enables the proposed method to achieve higher performance on seen categories without degrading the performance on unseen categories by involving contexts only when testing on seen categories and discarding it for unseen categories. We adopt this way for obtaining final scores.

| Context | | Seen Category | | | Unseen Category | | | | | | |
|---|---|---|---|---|---|---|---|---|---|---|---|
| | | 🪑 | 💡 | 🗄 | 🛍 | 🛏 | 🍶 | 🥣 | 🕐 | ⌨ | 🖥 |
| w/ | L1 | 62.7 | 68.9 | 24.7 | 38.1 | 14.4 | 57.5 | 71.3 | 57.4 | 33.9 | 70 |
| | L2 | 47.6 | 57.5 | 20.8 | - | 12.3 | - | - | - | 27.9 | - |
| | L3 | 41.5 | 44.5 | 19.7 | - | 10.9 | 34.6 | - | 25.6 | 19 | 60.7 |
| | Avg | **50.6** | **57** | **21.7** | 38.1 | **12.3** | 46.1 | 71.3 | 41.5 | 26.9 | 65.4 |
| w/o | L1 | 59.8 | 67 | 24.4 | 41.6 | 12 | 61.9 | 72.2 | 57.6 | 40.8 | 71.9 |
| | L2 | 44.6 | 55.2 | 19.2 | - | 10.6 | - | - | - | 32.6 | - |
| | L3 | 39.1 | 42.9 | 18.1 | - | 8.7 | 36.5 | - | 27.1 | 20.1 | 62.1 |
| | Avg | 47.8 | 55 | 20.6 | **41.6** | 10.4 | **49.2** | **72.2** | **42.4** | **31.2** | **67** |

Table 3: Quantitative evaluation of involving more context. w/ and w/o denote making decision with and without involving more context, respectively. Note that we only introduce more context in the late grouping process and the involved context is restricted in a very local region. The number is the mean recall of segmentation results. The L1, L2 and L3 refer to the three levels of segmentation defined in PartNet. Avg is the average among mean recall of three levels segmentation results.

## 5.6 COMPONENTS ANALYSIS

In our pipeline, we use the policy network to learn to pick pairs of sub-parts. It consists of the purity module and the rectification module, which process the unary information and binary information respectively. Here, we show the quantitative results and validate the effectiveness of these components. The results are listed in Table 4, where we train the model on the Chair, Lamp, Storage Furniture of level-3 annotations.

- **Purity Module:** The purity module takes the unary information (a grouped pair) as input and output the purity score. Similar to the objectness score Alexe et al. (2012) used in object detection, the purity score serves as the partness score to measure the quality of the proposal. We optimize the purity module by regressing the ground-truth purity scores and use such meaningful supervision to help learn the policy. The results of "no purity" row in Table 4 show the effectiveness of this module.
- **Rectification Module:** The rectification module is involved to rectify the failure cases for the purity network. Our experiments shows that without the rectification module, our decision process will easily converge to a trajectory that a pair of sub-part with unbalanced size will usually be chosen to group results in situations that one huge sub-part dominate the sub-part pool and bring in performance drop as shown in Table 4, the "no rectification" row. Please also refer to Appendix B to see some relating qualitative results.

| | Seen Category | | | Unseen Category | | | | | | |
|---|---|---|---|---|---|---|---|---|---|---|
| no purity | 38.6 | 36.8 | 5.6 | 30.3 | 7.0 | 29.4 | 63 | 21.1 | 10.9 | 53.5 |
| no rectification | 38.4 | 36.4 | 5.5 | 29.7 | 6.9 | 27.5 | 57.6 | 22.1 | 10.3 | 52.8 |
| full-model | 38.8 | 37.6 | 5.7 | 33.1 | 7.2 | 32.6 | 66.5 | 23.0 | 10.5 | 55.2 |

Table 4: Quantitative results of the components analysis. We train the models on the Chair, Lamp, Storage Furniture of level-3 annotations and test on the listed categories. The number is the mean recall of the most fine-grained annotations of each category.

## 6 CONCLUSION

In this paper, we introduced a data-driven iterative perceptual grouping pipeline for the task of zero-shot 3D shape part discovery. At the core of our method is to learn part-level features within part local contexts, in order to generalize the part discovery process to unseen novel categories. We conducted extensive evaluation and analysis of our method and presented thorough quantitative comparisons to four state-of-the-art shape segmentation algorithms. We demonstrated that our method successfully extracts locally-aware part knowledge from training categories and transfers the knowledge to unseen novel categories. Our method achieved the best performance over all four baseline methods on the PartNet dataset.

## 7 FUTURE WORK

There are several avenues for future research. Firstly, we formulate the grouping process as a series of contextual bandit problem, which greedily maximizes the defined score per step. Instead, we can do long-term planning to select pairs that maximize the expected future return in the later grouping process. Secondly, we assume all kinds of parts of unseen categories are included in the training data so that the network can generalize in the unseen categories. We may avoid such an assumption if there are few samples of the unseen categories. Assembling the information of those template samples, the model should be able to infer what kinds of novel parts are contained in the unseen categories. Furthermore, the experimental results in Section 5.5 indicates that involving more context can improve the transfer performance if the test categories and the training categories are similar. We may be able to detect such similarities, adaptively use more context for similar parts and improve the performance on both seen and unseen categories. Also, we currently only have the forward process (i.e., grouping) but lack the backward process, such as split operations. The backward process can fix errors accumulated in the grouping process. Finally, it would be interesting to apply this algorithm on the 2D domain to see whether the algorithm could help in the few-shot 2d object detection and segmentation.

## 8   ACKNOWLEDGEMENT

We thank Jiayuan Gu for the helpful discussion and support of code base and Yi Li for supporting in comparing GSPN. This work is in part supported by NSF grant IIS-1764078 and Qualcomm.

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

# A    SUB-PART PROPOSAL MODULE

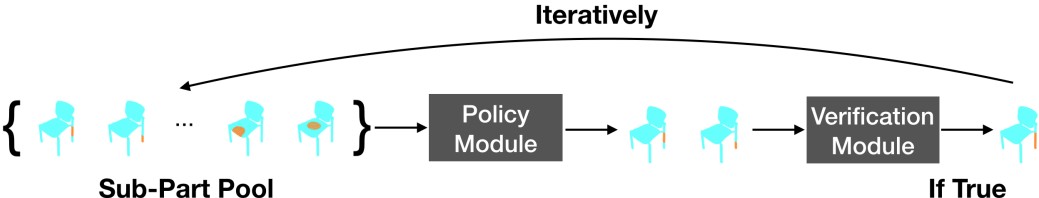

Figure 5: Overview of the proposed approach. The orange part is the point cloud of sub-parts; the blue part is the whole shape point cloud. We only feed sub-parts into our networks. In each iteration, we use a policy network to select a pair of sub-parts and send it to the verification network to verify whether we should group the selected pair of sub-parts. If yes, we group the selected pair of sub-parts into a larger sub-part, and put the grouped sub-part into the sub-part pool and delete the input pair from the pool. Otherwise, we will not consider this pair in the latter grouping process. We will iteratively do this process until no sub-part can further group each other. The remaining sub-parts in the final stage become a pool of part proposals for the input shape.

Given a shape represented as a point cloud, we first propose a pool of small superpixel-like (Ren & Malik, 2003) sub-parts (The orange part shown in Figure 5) as the building blocks. We employ furthest point sampling to sample $128$ seed points on each input shape. To capture the local part context, we extract PointNet (Qi et al., 2017a) features with $64$ points sampling within a local $0.04$-radius[2] neighborhood around each seed point. In the training phase, all the $64$ points will be sampled from the same instance. Then, we train a local PointNet segmentation network that takes as inputs $512$ points within a $0.2$-radius ball around every seed point and output a binary segmentation mask indicating a sub-part proposal. If the point belongs to the instance is the same as the $0.04$-radius ball, it will be classified into $1$. We call this module as the sub-part proposal module and illustrate it in Figure 6.

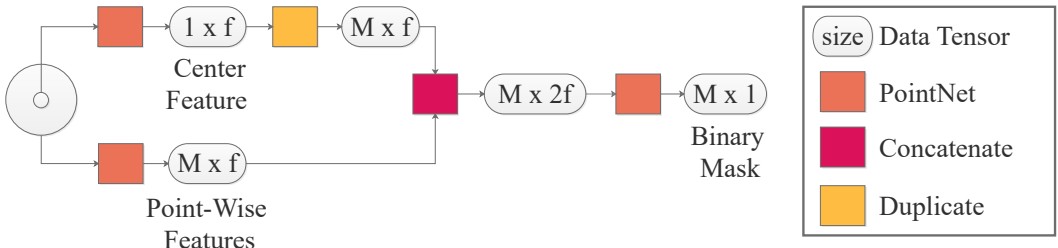

Figure 6: Learning-based sub-part proposal module.

In the inference phase, we can not guarantee the $64$ points sampled within a $0.04$-radius ball are all coming from the same part. However, in our experiments, we observe those sub-part proposals will have a low purity score due to the poor center feature extracted from the $64$ points across different parts. Also, even the center feature extraction is good, some sub-parts may also cover multiple parts in ground-truth. To obtain high-quality sub-parts, we remove the sub-parts whose purity score lower than $0.8$, and the remain sub-parts form our initial sub-part pool.

---

[2]All shape point clouds are normalized into a unit-radius sphere.

The input of this learning module is constrained in a local region, thus will not be affected by the global context. To validate the transferring performance of this module, we train the module on Chair, Storage Furniture, and Lamp of level-3 annotations and test on all categories with evaluating by the most fine-grained level annotations of each category. The results are listed in Table 5. Since the part patterns in Table are very similar to the patterns in Chair, we can see the zero-shot performance on Table is close to the performance on training categories. This phenomenon also aligns the analysis in Section 5.4.

| | Seen Category | | | Unseen Category | | | | | | | | |
|---|---|---|---|---|---|---|---|---|---|---|---|---|
| | | | | | | | | | | | | |
| PosAcc | 94.1 | 95.8 | 91.7 | 88.7 | 86.7 | 97.1 | 97.1 | 88.2 | 87.7 | 90.6 | 84.3 | 88.3 |
| NegAcc | 66.5 | 61.3 | 73.8 | 59.9 | 64.9 | 12.9 | 20.2 | 20.3 | 42 | 66 | 60.6 | 31.9 |
| | Unseen Category | | | | | | | | | | | |
| | | | | | | | | | | | | |
| PosAcc | 96.4 | 98.1 | 98.6 | 98.1 | 97 | 86.2 | 96.3 | 86.3 | 91.7 | 95.9 | 93.6 | 89.1 |
| NegAcc | 39.4 | 40.7 | 3.8 | 33.7 | 77.6 | 42.8 | 45.5 | 46.9 | 62.3 | 67.7 | 31.1 | 36.2 |

Table 5: Quantitative evaluation of the sub-part proposal module. PosAcc and NegAcc refer to positive accuracy and negative accuracy of the binary segmentation.

## B    RELATIVE SIZE VISUALIZATIONS

We involve the rectification module and learn the policy to pick pairs of sub-parts for grouping. The rectification module may bring several benefits. Here we demonstrate the effectiveness of the rectification module from one aspect that this module will encourage to pick equal size pairs of sub-parts. In our experiments, we found if we only follow the guidance of the purity network, our policy will tend to choose the pair comprising one big sub-part and one small sub-part. Like the descriptions in Section 4.1, the geometry of such unequal size pairs will be dominated by the big sub-part and thus raise the possibilities of errors. The rectification module can alleviate this situation and encourage the learned policy to choose more equal size pairs.

To evaluate this point, we define the relative size for the selected pairs. Given a pair of sub-parts $P_i$ and $P_j$, we define the relative size as $\frac{P_i}{P_j} + \frac{P_j}{P_i}$ where the smaller value means the size of the pair $P_i$ and $P_j$ are more equal, and the minimum value is 2. We train two models with and without the rectification module separately on Chair of level-3 annotations and test on Chair. We plot the relative size for the process of grouping and show some randomly sampled results in Figure 7. Every picture shows the grouping process for one shape, the x-axis is the iteration number of the process, the y-axis is the defined metric.

From the results, we can clearly see the rectification module helps to choose more equal size pairs. Only when it comes to the late stage, where the size of parts is various and it is hard to find size equal pairs, our policy will pick size unequal pairs. Therefore, the rectification module helps to prevent the trajectory converging to catastrophic cases in which larger sub-parts dominate the feature for purity score prediction and fail to predict the purity for the grouped sub-parts. Also, intuitively, the intermediate sub-parts generated during the grouping process may have various patterns and are irregular. This increases the burden of models to recognize, and the learned "equal-size selection" like rule may help to form regular intermediate sub-parts and alleviate this issue.

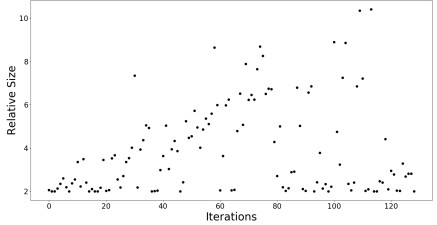 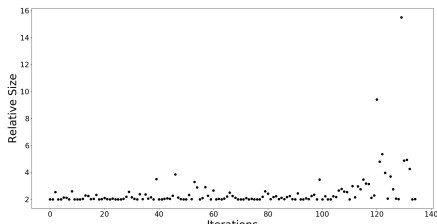

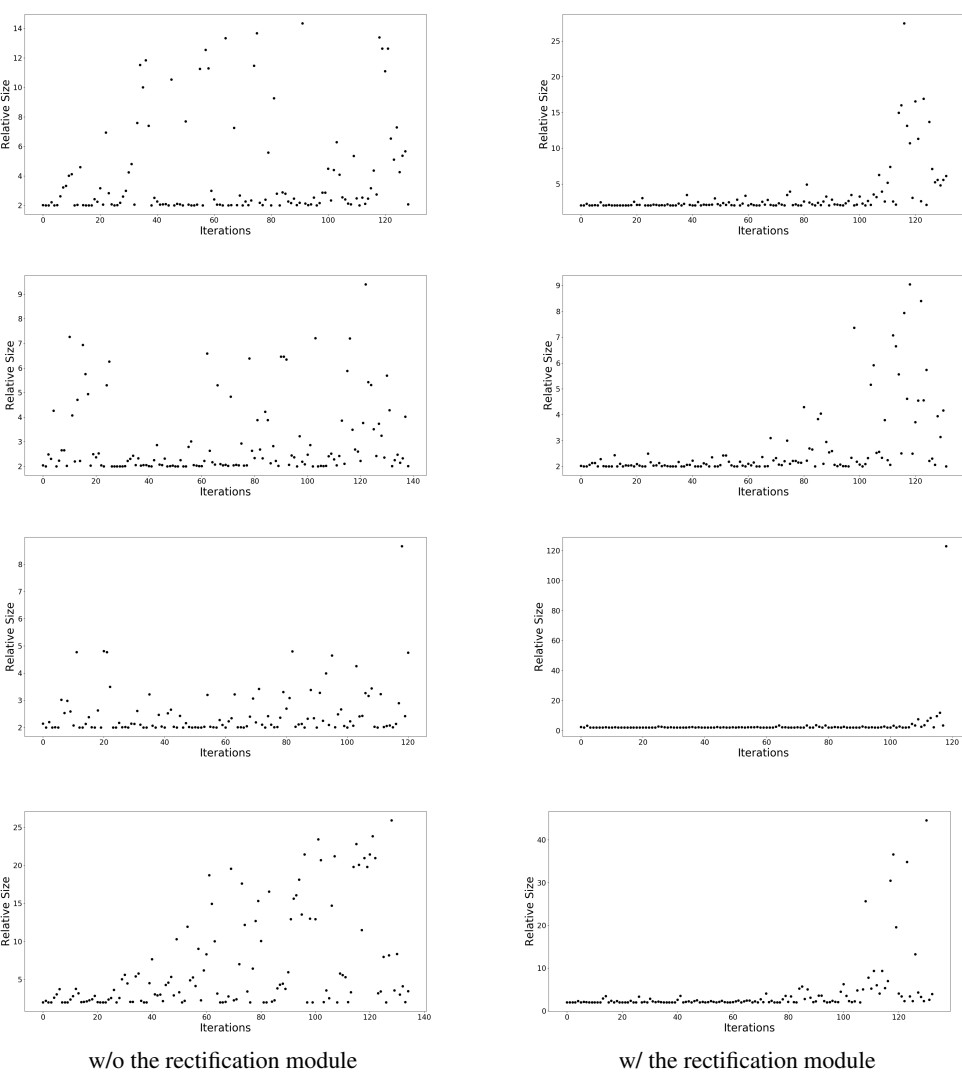

w/o the rectification module                                     w/ the rectification module

Figure 7: Qualitative results. Figures show the changes of the relative size of a pair of sub-parts during the grouping process. The X-axis is the iteration number and Y-axis is the relative size defined in Appendix B. Each row represents the grouping process for the same shape. The left column is the results without the rectification module.

## C   FULL EXPERIMENTS

### C.1   IMPLEMENTATION DETAILS

We will train our model on three categories (Chair, Lamp, and Storage Furniture) and test on all categories. For each method, we train three models corresponding to three levels of segmentation annotations for training categories. All the point clouds of shapes used in our experiments have 10000 points. For the compared baselines, the input is the whole shape point cloud, where the size is 10000. For the proposed method, the input is the points sampling from a sub-part, where the size is 1024. Therefore, the proposed method has advantages in GPU memory cost. Since our task does not need semantic labels, we remove the semantic segmentation loss for all the deep learning methods.

- **PartNet-InsSeg:** We follow the default settings and hyper-parameters described in the paper where the output instance number is 200, and loss weights are 1 except 0.1 for the

regularization term. We train the model for a total of 120 epochs with the learning rate starting from 0.001 and decaying 0.5 every 15 epochs. The whole training process will take 4 days on a single 1080Ti GPU.

- **SGPN:** Following the same experiment setting in Mo et al. (2019b), where the max group number is set to 200. The learning rate is 0.0001 initially and decays by a factor of 0.8 every 40000 iterations. It takes 3 days to train the network and 20 seconds to process each shape in the inference phase.

- **GSPN:** The maximum number of detection per shape is 200. The number of points of each generated instance is set to 512. NMS ( Non-Maximum Suppression) of threshold 0.8 is applied after proposal generation. As in Yi et al. (2019b), we train GSPN first for 20 epochs and then fine-tune it jointly with R-PointNet for another 20 epochs. The learning rate is initialized to 0.001 and decays by a factor of 0.7 every 10000 steps. Each stage of training takes 2 days respectively.

- **WCSeg:** This method requires out-ward normal as input which is lacked in the point clouds. In order to perform this method and compare it as fair as possible, we first generate out-ward normals for input point clouds as follows: a) we employ ball-pivoting Bernardini et al. (1999) to reconstruct surface for input point clouds. b) we keep one face fixed and re-orient the faces order coherently so that the generated face normal is all out-ward. c) we transfer the face normals back to the vertices' normals.

  As a traditional method, the performance is very sensitive to all hyper-parameters, we tune four parameters recommended in the paper by grid searching on seen categories and then test on unseen categories. More specifically, we randomly select 100 object instances for each of the three seen categories (i.e. Chair, Lamp, and Storage Furniture) as our grid search dataset. Then we conduct a grid search on the 300 instances regarding the parameters $(\theta_1, \theta_2, \theta_3, \sigma)$ in WCSeg. Based on the recommended parameters from the original paper Kaick et al. (2014), we apply relative shifts with the range of $[-20\%, +20\%]$ on each parameter to form $3^4 = 81$ sets of parameters. Among these parameters, we choose the set with the highest mean recall on fore-mentioned grid search dataset as the parameter for each level. We eventually select $(\theta_1 = 0.950, \theta_2 = 0.704, \theta_3 = 0.403, \sigma = 0.069)$ for Level-1, $(\theta_1 = 1.426, \theta_2 = 0.845, \theta_3 = 0.504, \sigma = 0.086)$ for Level-2, and $(\theta_1 = 1.188, \theta_2 = 0.563, \theta_3 = 0.504, \sigma = 0.069)$ for Level-3 in our experiments. We use the MATLAB code provided by the paper and perform it on our Intel(R) Xeon(R) Gold 6126 CPU cluster with 16 CPU cores used. For the inference, WCSeg takes about 2.2 minutes to process each shape per CPU core and about 4 days to finish testing over PartNet's part instance segmentation dataset.

- **Our:** For each shape, we first use the sub-part proposal module to generate sub-part proposals as described in Appendix A. For training the proposal module, we use batch size 12 and learning rate starting from 0.001 and decaying 0.5 every 15 epochs for a total of 120 epochs. After this, we will gain 128 proposals and then remove the proposals whose purity score is lower than 0.8. The rest proposals form our initial sub-parts pool. In the training phase, we can calculate the ground-truth purity score according to the annotations. In the inference phase, we will use the trained purity module to predict the purity score. To short the whole training time, we train the proposal module on level-3 annotations and use it for training the policy network and verification network on all three levels.

  After gaining the initial sub-part pool, we begin our grouping process. During the process, we will use the policy network comprising the purity module and the rectification module to pick the pair of sub-parts and use the verification network to determine whether we should group the pair. If a pair of sub-parts should be grouped, we will add the new grouped part into the sub-parts pool and remove the input two sub-parts from the pool. We will iteratively group the sub-parts until no more available pairs. So for each shape, we generate a trajectory and collect training data from the trajectory.

  In the training phase, for each iteration, we sample 64 pairs and calculate the policy score by using both the purity module and the rectification module. To accelerate the training process, we sample 10 pairs not 1 pair from the 64 pairs and send them to the verification network to determine whether we should group the pair of sub-parts. The 10 pairs comprise rank $n$ pairs and $10 - n$ random sampling pairs since we adapt epsilon-greedy strategy where start from involving 8 random sampling samples and decay the number with 1 in each epoch. The minimal number of random sampling pairs is 1. In the inference phase, for each iteration, we will calculate the policy score for all pairs and send the pair with the highest score to the verification network. Note that we use the prediction of the verification

network not the ground-truth annotations to determine whether we should group and generate the trajectory. According to ablation studies in Appendix 5.6, this on-policy manual is important for the final performance. For the level-3 model, we choose the pairs where two sub-parts are close to each other within the minimum distance $\ell_2 \leq 0.01$. For the level-1 and level-2 model, we will first choose the neighboring pairs and group them. When all neighboring pairs have been processed, we remove the neighboring constraints and group the pairs until no more available pairs.

We collect the training data from the trajectories and form the replay buffer for each module, where each replay buffer only can hold up to data from 4 trajectories. The size of the input point cloud to all three modules is 1024 by sampling from the sub-parts. In each iteration, we train all the modules on corresponding replay buffers. We train the modules for a total of 1600 iterations with the learning rate starting from 0.001 and decaying 0.5 every 150 epochs. The batch size for the purity module, the verification network is 128, for the rectification module is 2. The whole training process will take 4 days on a single 1080Ti GPU. For the inference, the method takes about 3 seconds to process each shape.

## C.2 FULL EXPERIMENTAL RESULTS

We present the full table including Mean Recall scores at all levels and the performance on seen categories in Table 6. We involve more context only on seen categories the same as the way presented in Appendix 5.5.

## C.3 MORE QUALITATIVE RESULTS

In this section, we list more qualitative results of GSPN, SGPN, WCSeg, PartNet-InsSeg, and our proposed methods for the zero-shot part discovery. We train the models on Chair, Lamp, and Storage Furniture of level-3 (the most fine-grained level) of PartNet Dataset and test on the other unseen categories. We also list the corresponding most fine-grained ground-truth annotations for reference (Some categories may only have the level-1 annotation). Note that the ground-truth annotation only provides one possible segmentation that satisfies category-specific human-defined semantic meanings.

| | Seen Category | | | SAvg | WSAvg | Unseen Category | | | | | | | | |
|---|---|---|---|---|---|---|---|---|---|---|---|---|---|---|
| | (lamp) | (faucet) | (shelf) | | | (bag) | (bed) | (bottle) | (bowl) | (clock) | (keyboard) | (display) | (tablet) | (earphone) |
| P1 | 73.9 | 63.8 | 27.6 | 55.1 | 61.9 | 18.2 | 14.7 | 49.6 | 73.5 | 33.4 | 37.3 | 43.2 | 42.4 | 24.2 |
| P2 | 50.3 | 47.7 | 21.9 | 40 | 43.6 | - | 7.8 | - | - | - | 32.2 | - | - | - |
| P3 | 41.8 | 39.3 | 20.7 | 33.9 | 36.7 | - | 6.6 | 31.8 | - | 27.1 | 18.4 | 44 | 21.8 | 8.7 |
| Avg | **55.3** | 50.3 | **23.4** | 43 | **47.4** | 18.2 | 9.7 | 40.7 | **73.5** | 30.3 | 29.3 | 43.6 | 32.1 | 16.5 |
| S1 | 57.1 | 56.2 | 13.6 | 42.3 | 47.5 | 21.4 | 10.7 | 57.6 | 53.3 | 37.5 | 13 | 38.4 | 44.1 | 43.1 |
| S2 | 38.2 | 42.1 | 11.4 | 30.6 | 33.2 | - | 6.3 | - | - | - | 7.9 | - | 23.7 | - |
| S3 | 31.3 | 34.4 | 9.4 | 25 | 27.2 | - | 4 | 35.8 | - | 17.9 | 5.2 | 31.2 | 19 | 7.9 |
| Avg | 42.2 | 44.2 | 11.5 | 32.6 | 36 | 21.4 | 7 | 46.7 | 53.3 | 27.7 | 8.7 | 34.8 | 28.9 | 25.5 |
| G1 | 56.7 | 57.8 | 17.7 | 44.1 | 48.5 | 34.4 | 17.2 | 56 | 72.8 | 55.6 | 53.5 | 63.7 | 55.6 | 46.9 |
| G2 | 35.2 | 42 | 13.5 | 30.2 | 31.9 | - | 4.6 | - | - | - | 40.5 | - | 30.2 | - |
| G3 | 27.1 | 31.2 | 12.1 | 23.5 | 24.7 | - | 3.4 | 37.8 | - | 25.6 | 27.8 | 51.9 | 24.2 | 9.9 |
| Avg | 39.7 | 43.7 | 14.4 | 32.6 | 35 | 34.4 | 8.4 | 46.9 | 72.8 | 40.6 | **40.6** | 57.8 | 36.7 | 28.4 |
| W1 | 31.3 | 57.7 | 5.5 | 31.5 | 31 | 41.9 | 10.9 | 67 | 69.3 | 43.8 | 46.5 | 61.3 | 42.9 | 48.6 |
| W2 | 34.3 | 59.3 | 1.9 | 31.8 | 32.3 | - | 8.2 | - | - | - | 20.8 | - | 25.2 | - |
| W3 | 33.7 | 53.4 | 2.2 | 29.8 | 30.9 | - | 6.8 | 45.6 | - | 24.5 | 15.6 | 58 | 22.5 | 26 |
| Avg | 33.1 | 56.8 | 3.2 | 31 | 31.4 | **41.9** | 8.6 | **56.3** | 69.3 | 34.2 | 27.6 | 59.7 | 30.2 | **37.3** |
| O1 | 62.7 | 68.9 | 24.7 | 52.1 | 55.7 | 41.6 | 12 | 61.9 | 72.2 | 57.6 | 40.8 | 71.9 | 55.5 | 54.8 |
| O2 | 47.6 | 57.5 | 20.8 | 42 | 43.8 | - | 10.6 | - | - | - | 32.6 | - | 30.4 | - |
| O3 | 41.5 | 44.5 | 19.7 | 35.2 | 37.4 | - | 8.7 | 36.5 | - | 27.1 | 20.1 | 62.1 | 25.6 | 11.4 |
| Avg | 50.6 | **57** | 21.7 | **43.1** | 45.6 | 41.6 | **10.4** | 49.2 | 72.2 | **42.4** | 31.2 | **67** | **37.2** | 33.1 |

| | Unseen Category | | | | | | | | | | | | UAvg | WUAvg |
|---|---|---|---|---|---|---|---|---|---|---|---|---|---|---|
| | (faucet) | (hat) | (keyboard) | (pen) | (laptop) | (tablet) | (mug) | (fridge) | (scissors) | (table) | (trash) | (vase) | | |
| P1 | 19.4 | 52.5 | 0.4 | 43.2 | 82.1 | 42 | 33 | 31.6 | 0.8 | 56 | 21.1 | 38 | 36 | 47.8 |
| P2 | - | - | - | - | - | 28.5 | - | 25.4 | - | 32.4 | - | - | 25.3 | 31.7 |
| P3 | 13.8 | - | - | 23.9 | - | 18.3 | - | 18 | - | 28.4 | 3.2 | 35.5 | 21.4 | 27.5 |
| Avg | 16.6 | **52.5** | 0.4 | 33.6 | 82.1 | 29.6 | 33 | 25 | 0.8 | 38.9 | 12.2 | 36.8 | 31.2 | 35.7 |
| S1 | 23.3 | 37 | 0.4 | 39.3 | 67.3 | 11.1 | 13.3 | 7.5 | 6.4 | 48.2 | 12.7 | 28.6 | 29.2 | 41 |
| S2 | - | - | - | - | - | 7.1 | - | 5.4 | - | 29.4 | - | - | 13.3 | 27.3 |
| S3 | 16.6 | - | - | 22.7 | - | 3.4 | - | 4.9 | - | 26.7 | 2.8 | 26.3 | 16 | 24.1 |
| Avg | 20 | 37 | 0.4 | 31 | 67.3 | 7.2 | 13.3 | 5.9 | 6.4 | 34.8 | 7.8 | 27.5 | 24.4 | 30.8 |
| G1 | 32.5 | 31.7 | 0.4 | 25.6 | 92.9 | 62.3 | 40.6 | 41.4 | 3.7 | 49.9 | 20.8 | 42.4 | 42.9 | 48.6 |
| G2 | - | - | - | - | - | 34.6 | - | 21.4 | - | 28.8 | - | - | 26.7 | 28.7 |
| G3 | 18 | - | - | 12.2 | - | 20.7 | - | 13.4 | - | 25 | 2.2 | 40.3 | 22.3 | 26.6 |
| Avg | 25.3 | 31.7 | 0.4 | 18.9 | 92.9 | **39.2** | 40.6 | 25.4 | 3.7 | 34.6 | 11.5 | 41.4 | 34.9 | 34.6 |
| W1 | 48.4 | 48.7 | 0.3 | 64.7 | 64.8 | 54.5 | 46 | 36.8 | 39 | 36 | 21.7 | 29.7 | 43.9 | 40.8 |
| W2 | - | - | - | - | - | 22 | - | 13.1 | - | 30.7 | - | - | 20 | 29.4 |
| W3 | 48 | - | - | 55.6 | - | 15.8 | - | 8.6 | - | 27.4 | 2.9 | 28.3 | 27.6 | 30.2 |
| Avg | **48.2** | 48.7 | 0.3 | **60.1** | 64.8 | 30.8 | 46 | 19.5 | **39** | 31.4 | 12.3 | 29 | 37.9 | 33.5 |
| O1 | 37.2 | 34.1 | 0.4 | 54.2 | 96.6 | 55.6 | 48.2 | 42.3 | 16.7 | 61.5 | 22.5 | 44.7 | 46.8 | 56.8 |
| O2 | - | - | - | - | - | 29.2 | - | 22.5 | - | 37.8 | - | - | 27.2 | 36.5 |
| O3 | 24.6 | - | - | 34 | - | 18.2 | - | 15 | - | 33.1 | 3.4 | 41.5 | 25.8 | 33.1 |
| Avg | 30.9 | 34.1 | 0.4 | 44.1 | **96.6** | 34.3 | **48.2** | **26.6** | 16.7 | **44.1** | **13** | **43.1** | **38.9** | **42.1** |

Table 6: Quantitative Evaluation. Algorithm P, S, G, W, O refer to PartNet-InsSeg, SGPN, GSPN, WCSeg and Ours, respectively. The number 1, 2 and 3 refer to the three levels of segmentation defined in PartNet. We put short lines for the levels that are not defined. Avg is the average among mean recall of three levels segmentation results in PartNet. SAvg and WSAvg are average among seen categories and weighted average among seen categories over shape numbers, respectively. UAvg and WUAvg are average among unseen categories and the weighted average among unseen categories over shape numbers, respectively.

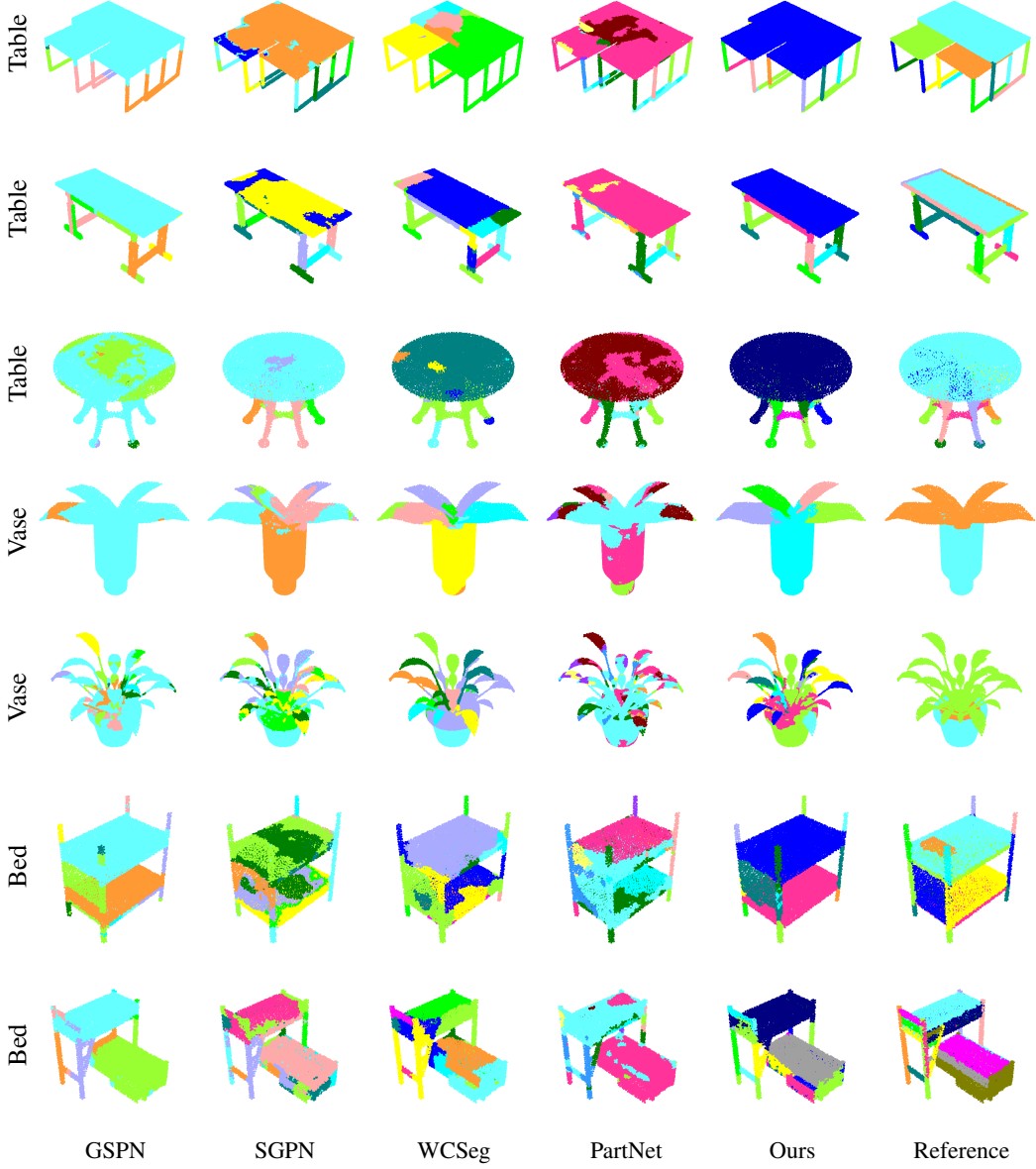

GSPN      SGPN      WCSeg      PartNet      Ours      Reference

Figure 8: We train the models on Chair, Lamp, and Storage Furniture of level-3 (the most fine-grained level) of PartNet Dataset and test on the other unseen categories listed in the right. The rightmost column is the most fine-grained ground-truth annotations for reference. Note that the ground-truth annotation only provides one possible segmentation that satisfies category-specific human-defined semantic meanings.

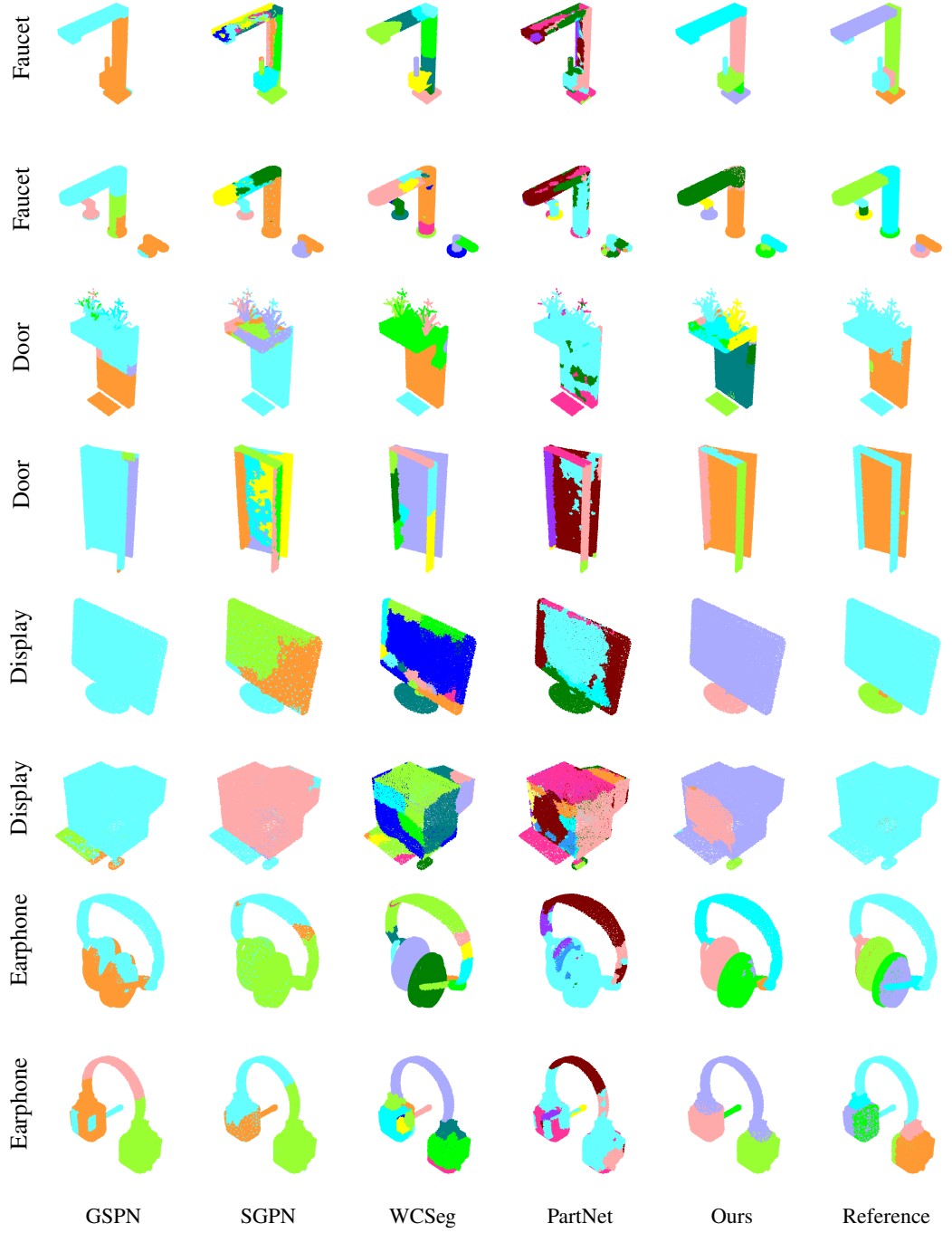

Figure 9: We train the models on Chair, Lamp, and Storage Furniture of level-3 (the most fine-grained level) of PartNet Dataset and test on the other unseen categories listed in the right. The rightmost column is the most fine-grained ground-truth annotations for reference. Note that the ground-truth annotation only provides one possible segmentation that satisfies category-specific human-defined semantic meanings.

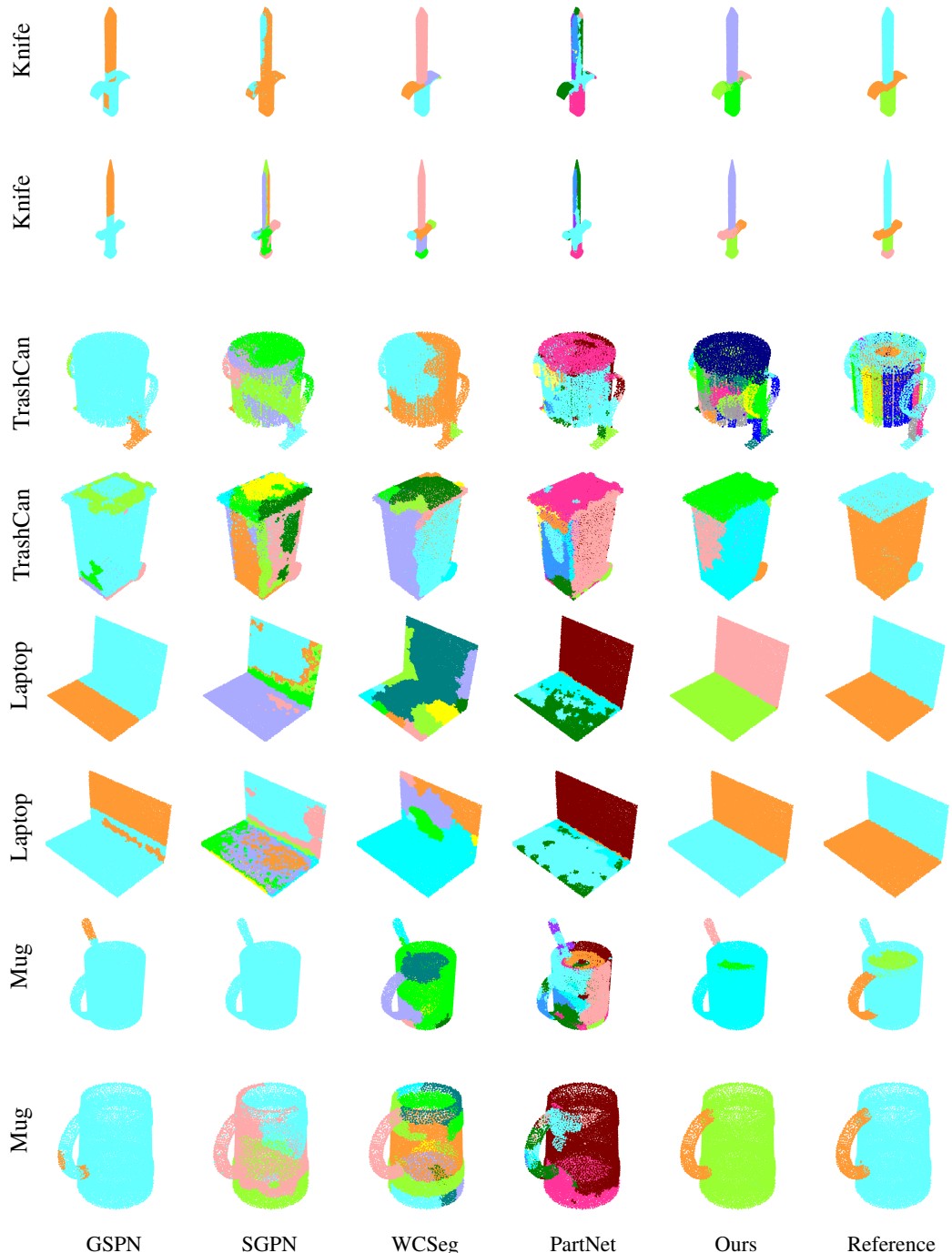

GSPN      SGPN      WCSeg      PartNet      Ours      Reference

Figure 10: We train the models on Chair, Lamp, and Storage Furniture of level-3 (the most fine-grained level) of PartNet Dataset and test on the other unseen categories listed in the right. The rightmost column is the most fine-grained ground-truth annotations for reference. Note that the ground-truth annotation only provides one possible segmentation that satisfies category-specific human-defined semantic meanings.

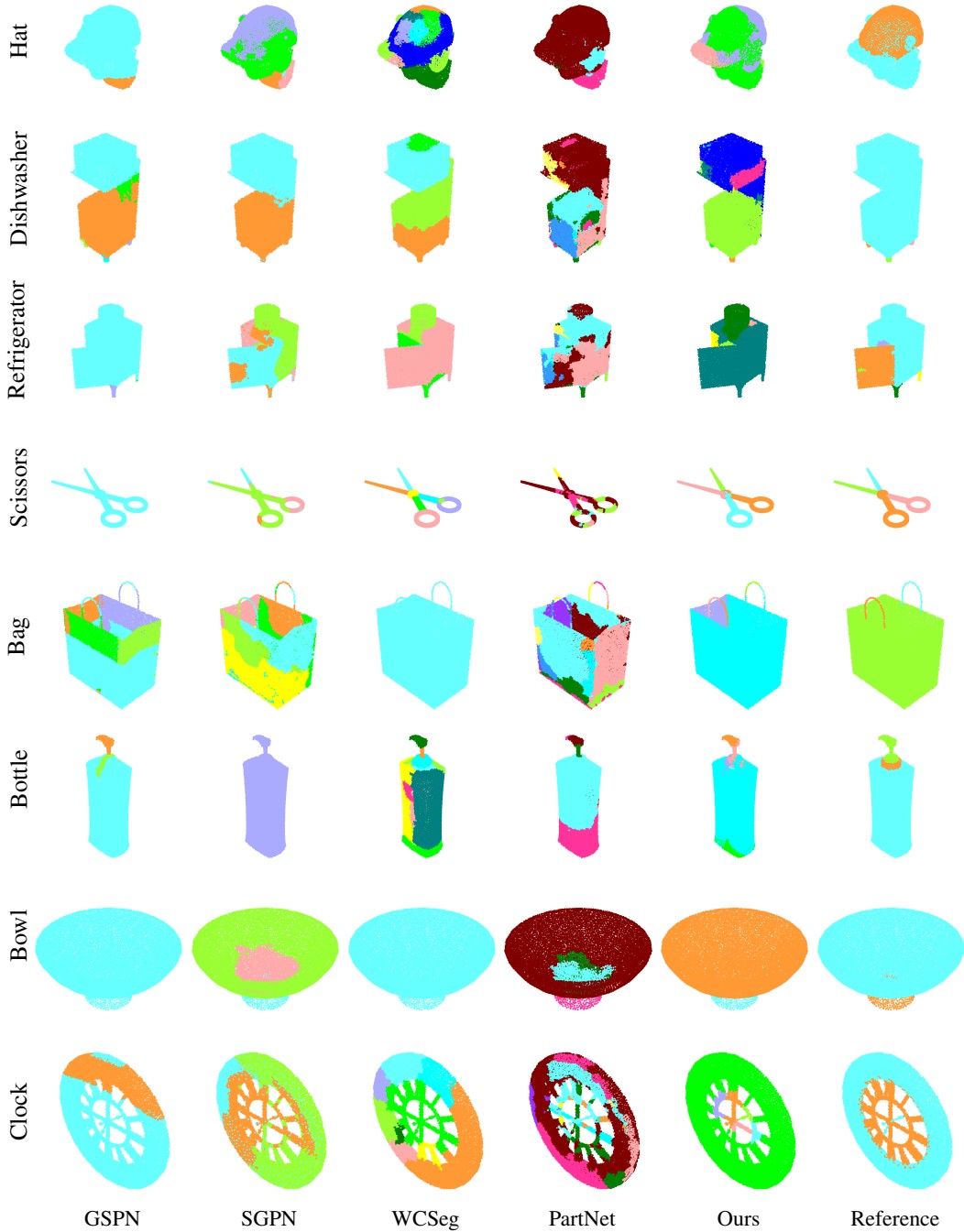

Figure 11: We train the models on Chair, Lamp, and Storage Furniture of level-3 (the most fine-grained level) of PartNet Dataset and test on the other unseen categories listed in the right. The rightmost column is the most fine-grained ground-truth annotations for reference. Note that the ground-truth annotation only provides one possible segmentation that satisfies category-specific human-defined semantic meanings.

