# OpenReview forum: "Learning to Group: A Bottom-Up Framework for 3D Part Discovery in Unseen Categories"
_ICLR.cc/2020/Conference — Accept (Poster)_

### Official Review · AnonReviewer3 · 2019-10-22
**Official Blind Review #3**

**Rating:** 8

**Review:**

This paper describes a method for segmenting 3D point clouds of objects into component parts, with a focus on generalizing part groupings to novel object categories unseen during training.  In order to improve generalization, the paper argues for limiting the influence of global context, and therefore seeks to build compact parts in a bottom-up fashion by iterative merging of superpixel-like point subsets.  This is achieved by defining a RL merge policy, using merge and termination scores formed by a combination of explicitly trained part purity (each part should comprise one true part), and policy-trained pair comparison network.  The system is evaluated using PartNet, using three categories for training and the rest for testing, showing strong performance relative to baselines.

The system is described well, and shows good performance on a nicely motivated task.  A few more ablations would have been nice to see (in questions below), as might more qualitative results.  Overall, the method is presented and evaluated convincingly.


Questions:

*  What is the effect of the purity score regression?  Since the policy network is trained using a pair-comparison module anyway, what happens if the explicit purity score supervision is removed?

* What if the "rectifier" module is made larger (with or without purity module), e.g. the same size as the termination network?  Does this improve or overfit to the training categories?

* Sec 5.3 mentions "segmentation levels for different categories may not share consistent part granularity ....  Thus, ... we train three networks corresponding to three levels of segmentation for training categories".  While it makes sense to have three networks for the three levels (each have different termination points, and perhaps even merge paths), I don't see how this follows from the levels being inconsistent between categories.  In fact, it seems just the opposite, that if the levels are inconsistent, this could pose a problem when a part at one level for one category is "missing" from the other category, due to level numbers not coinciding.  Or, is this actually not a problem because on the three training categories selected, the levels are in fact consistent?

* Can termination be integrated into the policy network or policy itself?


A couple typos I noticed:

p.5 "In consequences," --> "As a consequence,"
p.11 "in-balanced" --> "unbalanced"


**Experience Assessment:**

I have read many papers in this area.

**Review Assessment: Checking Correctness Of Derivations And Theory:**

I assessed the sensibility of the derivations and theory.

**Review Assessment: Checking Correctness Of Experiments:**

I assessed the sensibility of the experiments.

**Review Assessment: Thoroughness In Paper Reading:**

I read the paper at least twice and used my best judgement in assessing the paper.

---

> ### Author Response · Authors · 2019-11-11
> **Response to Reviewer #3**
>
> We thank Reviewer #3 for the comments and suggestions. The suggestions are helpful in further improving our work.
>
> Here are the answers to the questions and concerns:
>
>
> [Regarding the purity score and the purity module]
> Similar to the objectness score used in object detection, the purity score serves as the partness score to measure the quality of the sub-part. The purity score is higher, the sub-part is more likely to only cover one part in ground-truth. We will use this score to measure the quality of our initial sub-part proposals and remove the low-quality subparts to form our initial sub-parts pool. We add more related descriptions in Appendix A and C.1.
>
> For the policy network, we use the purity module to process unary information and the rectification module to process binary information. While the purity score as a unary term could be learned through the policy gradient, we observe that we can give direct supervision, which is effective. In revision, we add related ablation results in Appendix B.2 and justify that the purity module helps to learn the policy.
>
>
>
> [Regarding making the rectification module larger]
> Larger networks have larger capacities but higher risks to overfit to training data. Since our approach purely exploits local context, larger networks may have less impact on the overfitting issue for our method compared to those baselines with inputting the global context. We enlarge the rectification module and gain improvements in seen categories. For the unseen categories which have similar part patterns with the training categories, we obtain some improvements. But, for the unseen categories (e.g. scissors) which have relatively large different part patterns with the training categories (chair, storage furniture, lamp), we observe inconsistent improvements or declinations. Thanks for the suggestion. We will study this point thoroughly, and include it in revision.
>
> Please also refer to Appendix B.1 to see the related ablation studies about the effects of involving more context on both seen and unseen categories.
>
>
>
> [Regarding the levels being inconsistent between categories]
> Thanks for pointing this out. The mentioned statements are not clear, and we have fixed it in revision. The segmentation levels for different categories may not share consistent part granularity, which is the reason that we gather together the part proposals predicted by networks at all three levels as a joint pool of proposals for evaluation on unseen categories. The levels of different categories may not correspond exactly; however, the joint part proposals can cover multiple levels of parts for unseen categories. Our three training categories have several thousands of models per category, thus providing a large variety of parts at different granularities for learning.
>
>
>
> [Regarding integrating the termination module into the policy module]
> In our pipeline, we will use the policy network to pick the pair of sub-parts and use the termination network to determine whether we should group the pair. The termination module is the basic building block of our pipeline. We noticed that the name of “termination module” may have confused readers, so we would rename it as “verification module” and made related descriptions clearer in revision. Also, we would like to point out that the termination module will focus on the samples selected by the policy module. This cascade structure serves as a kind of hard example mining and will improve the performance.

---

### Official Review · AnonReviewer1 · 2019-10-25
**Official Blind Review #1**

**Rating:** 6

**Review:**

This paper proposes a method for part segmentation in object pointclouds. The method is to (1) break the object into superpixel-like subparts (without semantic meaning yet), then (2) score pairs of parts on their mergeability, (3) greedily merge the best pair, and repeat. The scoring has a unary component (called a "purity" module), and a pairwise component (called a "rectification" module); the unary component determines if the joined pointcloud of two sub-parts appears part-like, and the pairwise component determines if the features of the two sub-parts appear compatible. These components are implemented as pointnets/MLPs. Finally there is a termination module, which sigmoid-scores part pairs on whether they should actually merge (and the algorithm continue), or not (and we stop). The purity and termination modules are trained supervised, to mimic intersection-like and mergeability scores, and the rectification module with a "reward" which is another mergeability score (coming from GT and the purity module).

The method is interesting for being (1) iterative, and (2) driven by purely local cues. The iterative approach, with small networks doing the work, is a nice relief from the giant-network baselines (such as PartNet-InsSeg) that take the entire pointcloud as input and produce all instance segmentations directly. Also, whereas most works try to maximize the amount of contextual input to the learning modules, this work makes the (almost certainly correct) observation that the smaller the contextual input, the smaller the risk for overfitting. This is a bit like making the approach "convolutional", in the sense that the same few parameters are used repeatedly over space (and in this case, also repeated over scale). The design of the local modules makes sense, although I would prefer they be called unary/pairwise instead of purity/rectification, and the RL training procedure looks reasonable also.

I am not totally clear on how the termination module actually comes into play. From the name, it sounds like this network would output 1 when the algorithm should terminate, but in its usage, it seems to output 1 when the best-scored pair should be merged. So then, does the algorithm terminate when this module decides to NOT merge the best-scored pair? This sounds like it bears great risk of early stopping. I would appreciate some clarification on this.

The abstract says that locality "guarantees the generalizability to novel categories". This is an overstatement, since "guarantees" implies some theoretical proof, and also since the paper's own results (in Table 1 and 3) indicate that cross-category generalization is far from addressed, and depends partly on the categories used in training (shown in Table 2).

I assume that this method has (or at least can have) far fewer parameters than the baselines, since the components never need to learn broad contextual priors. Can the authors clarify and elaborate on this please? If you can show that your method has far fewer parameters than the baselines, it would improve the paper I think.

Can the authors please provide some statistics on the earliest stage of the method, where superpixel-like parts are proposed? How many proposals, and how many pairs does this make, and how slowly do the main modules proceed through these pairs?

Is there a missing step that makes the part selection non-random? It seems like many of the pairs can be rejected outright early on, such as ones whose centroids exceed some distance threshold in 3D.

**Experience Assessment:**

I have read many papers in this area.

**Review Assessment: Checking Correctness Of Derivations And Theory:**

I assessed the sensibility of the derivations and theory.

**Review Assessment: Checking Correctness Of Experiments:**

I assessed the sensibility of the experiments.

**Review Assessment: Thoroughness In Paper Reading:**

I read the paper thoroughly.

---

> ### Author Response · Authors · 2019-11-11
> **Response to Reviewer #1**
>
> We greatly appreciate Reviewer #1 for the analysis, which precisely states the contributions of this paper. We have added more details and made descriptions clearer in revision.
>
> Here are the answers to the questions and concerns:
>
>
> [Regarding the naming of modules, especially the “termination module”]
> Thanks for pointing this out. After reading the comments from all reviewers, we also feel that renaming some modules may help to clarify confusion.
>
> Since the policy scores sum to one overall pairs of sub-parts, there is no explicit signal from the policy network whether the pair should be grouped.  We therefore introduce the termination module to verify whether we should group the pair of sub-parts, selected based on the score from the policy module. In fact, this module will verify pairs of merge candidates ranked by their scores from the policy module. The first pair that passes the verification will be grouped. Only if no pairs pass the verification the whole algorithm will terminate. Based on the functionality, we would rename this module as “verification module”. The verification module complements the policy module, which needs to recognize so many samples. The pipeline thus can be viewed as a kind of hard example mining. We have made the related descriptions clearer in both Section 4.1 and Appendix C.1.
>
> Currently, we named the modules according to their functionalities. We thanks for the suggestion and will seriously consider using unary/pairwise(binary) naming the purity/rectification modules which indicate the type of information input into the module.
>
>
>
> [Regarding the overstatement]
> Thanks for the suggestion. We have articulated this statement more precisely in revision. The phrase “guarantees the generalizability” is replaced by a milder one “encourages the generalizability”.
>
>
>
> [Regarding the proposed model has fewer parameters than baselines]
> Thanks for pointing this out. The number of parameters for different methods is listed below:
>
> --------------------------
> PartNet: 1.93e+06
> SGPN:   1.55e+06
> GSPN: 14.80e+06 (Shape Proposal Net: 13.86e+06)
> Our:       0.64e+06
> --------------------------
>
> Currently, our model does have fewer parameters than compared learning methods. But, we would like to point out that it will slightly degrade the performance on both seen and unseen categories if half the width of the model. Our intuition is that the intermediate sub-parts generated during the grouping process may have various patterns and are irregular. This increases the burden of models to recognize, and we widen the network can alleviate this situation. This intuition is also one of the motivations why we introduce the RL to learn to select the pairs. We want to use the policy network to help form more regular intermediate sub-parts during the grouping process. The size-equal rule learned by our policy network is a positive signal on this point. Please also refer to Appendix B.3 and see some related qualitative results.
>
> Besides, the input for our modules is not the whole shape point cloud, but sampling points of sub-parts from the shape. In our experiments, the size of input point clouds for our method is 1024, while for compared baselines are 10000. So the proposed method has advantages in GPU memory cost.
>
>
>
> [Regarding more statistics on the earliest stage of the method]
> Thanks for pointing this out, and we have added more details and statistics in Appendix A and C.1.
>
> When we train on Chair and test on Chair of the level-3 annotation, the average number of initial proposals on is 124 and the average number of pairs for the initial pool is 658. The number of valid pairs will decrease quickly as the grouping process going, and we usually have a total of 137 iterations. When employing the model on 1080Ti, it will cost 3s for processing one shape.
>
>
>
> [Regarding the part selection]
> Yes, we adopted similarly mentioned conditions that two sub-parts of a pair are constrained to be close to each other at the early grouping stage. Please refer to Appendix C.1 to see more implementation details.

---

> > ### Comment · AnonReviewer1 · 2019-11-15
> > **Thank you**
> >
> > This is helpful.
> >
> > Having fewer parameters and using less GPU memory are nice attributes of your model, which I encourage you to mention in the paper. (I have not noticed this appear in the draft so far.)
> >
> > I did not understand this sentence from your response: "it will slightly degrade the performance on both seen and unseen categories if half the width of the model". Could you please say that another way? Perhaps a word got deleted accidentally.

---

> > > ### Author Response · Authors · 2019-11-15
> > > **Response**
> > >
> > > Thanks for the updates!
> > >
> > > We also believe that the two points will strengthen the contribution of the proposed method, and include them in Appendix C.1. Thank you for the valuable suggestion!
> > >
> > > Sorry for the confusion. What we want to point out is that the capacity of the network also matters the performance of our method. If we cut the width of the model by half, the performance of both seen and unseen categories will drop slightly. Our intuition about this point is that the intermediate sub-parts generated during the grouping process may have various patterns and are irregular. Larger capacity will help the model recognize the various patterns. This intuition is also one of our motivations to introduce RL for learning to select pairs of sub-parts.

---

### Official Review · AnonReviewer2 · 2019-10-27
**Official Blind Review #2**

**Rating:** 3

**Review:**

This paper studies the problem of part segmentation in objects represented as a point cloud. The main novelty is in the fact that the proposed method uses a bottom-up iterative merging framework inspired by perceptual grouping and finds that it transfers better to unseen categories. In zero-shot transfer experiments, the proposed method performs better than all four other baselines compared; but is worse than Mo et al. (2019) in known categories.

The paper hypothesizes that top-down approaches do not generalizes well to new categories because they end up overfitting to the global context. While this is reasonable, I find that the experiments are not sufficient to validate this claim (please see questions below). Evaluation on unseen object categories is an underexplored topic, and the paper is generally well written. I think the submission can be an above-threshold paper if the questions are addressed.

- I’d like to see some evidence for the claim that classic segmentation methods "can perform much better for unseen object classes" (last paragraph of page 1), and see how the proposed method compares to those baselines.

- If my understanding of Table 3 is correct, "PartNet-InsSeg" (Mo et al. 2019) is a top-down approach yet it performs better than SGPN which is a bottom-up grouping method (as summarized on page 7) in novel categories. If so, can it be explained in a way that is consistent with the paper's findings?

- Table 4 shows some ablation study in an attempt to justify the proposed design, but I think it should be more thorough. e.g. it is not immediately obvious why the authors did not included a baseline that consists only of the rectification module with a termination threshold (seems like the most basic design that doesn't have the large-part bias or explicitly require a termination module).



Typos:

psilon-greedy   (page 6 paragraph 2)
backpropogation  (page 6 under training losses)
In consequences (page 5 under termination network)
epilson  (page 5, under network training)

**Experience Assessment:**

I have read many papers in this area.

**Review Assessment: Checking Correctness Of Derivations And Theory:**

I assessed the sensibility of the derivations and theory.

**Review Assessment: Checking Correctness Of Experiments:**

I carefully checked the experiments.

**Review Assessment: Thoroughness In Paper Reading:**

I read the paper at least twice and used my best judgement in assessing the paper.

---

> ### Author Response · Authors · 2019-11-11
> **Response to Reviewer #2**
>
> We thank Reviewer #2 for the feedback and suggestions. The suggestions are helpful, and we are open to further discussions.
>
> From the comments, we infer that Reviewer #2 assumes our claim to be that top-down approaches perform worse than bottom-up approaches in terms of generalization abilities. This is not exactly our view. Here, we precisely lay out our argument: Using features with the global context may hurt part segmentation performance in unseen categories. In most top-down pipelines and some bottom-up pipelines, the features extracted for each point to be fed to the classifier would include the global context. This point will be further discussed when addressing specific concerns.
>
>
> [Regarding “PartNet-InsSeg” outperforms “SGPN” in novel categories]
> Both “PartNet-InsSeg” (top-down) and the “SGPN” (bottom-up) involve global context to learn point features and make decisions, thus give inferior segmentation results on unseen categories. This is consistent with our conclusions. We are happy to make this point crystally clear in the revised version.
>
>
>
> [Regarding the performance of the tradition segmentation methods and the proposed method]
> WCseg is one of the most feasible traditional segmentation methods, whose results are provided in Table 1. Compared to the learning-based methods, It champions 6 out of 21 unseen categories. Also, we have added more qualitative results to Appendix C.3, which demonstrates the performance of both the traditional segmentation method and the proposed method.
>
>
>
> [Regarding the ablation studies]
> Thanks for pointing this out, and we made the ablation studies more thorough in revision, including the effects of involving more context on both seen and unseen categories, more components analysis, and qualitative results of the rectification module.  Please refer to Appendix B for details.
>
> Since the policy scores sum to one overall pairs of sub-parts, there is no explicit signal from the policy network whether the pair should be grouped.  We therefore introduce the termination module to verify whether we should group the pair of sub-parts, selected based on the score from the policy module. We noticed that the name of “termination module” may have confused reviewers, so we would rename it as “verification module”. Also, there is indeed a cascaded structure where the termination module will focus on the samples selected by the policy module. This serves as a kind of hard example mining and complements the policy module, which needs to recognize so many samples. We will make the related descriptions clearer in revision.
>
>
>
> [Regarding the proposed method performs worse than Mo et al. (2019) in seen categories]
> With involved limited context only for seen categories, our proposed method further improves the performance in seen categories. Please refer to Table 1,6 for new results and Appendix B.1 for details.

---

### Decision · Program_Chairs · 2019-12-19

**Decision:**

Accept (Poster)

**Comment:**

This paper presents and evaluates a technique for unsupervised object part discovery in 3d -- i.e. grouping points of a point cloud into coherent parts for an object that has not been seen before. The paper received 3 reviews from experts working in this area. R1 recommended Weak Accept, and identified some specific technical questions for the authors to address in the response (which the authors provided and R1 seemed satisfied). R2 recommends Weak Reject, and indicates an overall positive view of the paper but felt the experimental results were somewhat weak and posed several specific questions to the reviewers. The authors' response convincingly addressed these questions. R3 recommends Accept, but suggests some additional qualitative examples and ablation studies. The author response again addresses these. Overall, the reviews indicate that this is a good paper with some specific questions and concerns that can be addressed; the AC thus recommends a (Weak) Accept based on the reviews and author responses.